# The Association between Vitamin D Status and Autism Spectrum Disorder (ASD): A Systematic Review and Meta-Analysis

**DOI:** 10.3390/nu13010086

**Published:** 2020-12-29

**Authors:** Zuqun Wang, Rui Ding, Juan Wang

**Affiliations:** 1Department of Biomedical Informatics, School of Basic Medical Sciences, Peking University, Beijing 100191, China; wzq_quince@bjmu.edu.cn (Z.W.); dingrui@hsc.pku.edu.cn (R.D.); 2Autism Research Center, Peking University Health Science Center, Beijing 100191, China

**Keywords:** autism spectrum disorder, vitamin D, maternal and neonatal vitamin D, systematic review, meta-analysis

## Abstract

The association between vitamin D status and autism spectrum disorder (ASD) is well-investigated but remains to be elucidated. We quantitatively combined relevant studies to estimate whether vitamin D status was related to ASD in this work. PubMed, EMBASE, Web of Science, and the Cochrane Library were searched to include eligible studies. A random-effects model was applied to pool overall estimates of vitamin D concentration or odds ratio (OR) for ASD. In total, 34 publications involving 20,580 participants were identified in this present study. Meta-analysis of 24 case–control studies demonstrated that children and adolescents with ASD had significantly lower vitamin D concentration than that of the control group (mean difference (MD): −7.46 ng/mL, 95% confidence interval (CI): −10.26; −4.66 ng/mL, *p* < 0.0001, I^2^ = 98%). Quantitative integration of 10 case–control studies reporting OR revealed that lower vitamin D was associated with higher risk of ASD (OR: 5.23, 95% CI: 3.13; 8.73, *p* < 0.0001, I^2^ = 78.2%). Analysis of 15 case–control studies barring data from previous meta-analysis reached a similar result with that of the meta-analysis of 24 case–control studies (MD: −6.2, 95% CI: −9.62; −2.78, *p* = 0.0004, I^2^ = 96.8%), which confirmed the association. Furthermore, meta-analysis of maternal and neonatal vitamin D showed a trend of decreased early-life vitamin D concentration in the ASD group (MD: −3.15, 95% CI: −6.57; 0.26, *p* = 0.07, I^2^ = 99%). Meta-analysis of prospective studies suggested that children with reduced maternal or neonatal vitamin D had 54% higher likelihood of developing ASD (OR: 1.54, 95% CI: 1.12; 2.10, *p* = 0.0071, I^2^ = 81.2%). These analyses indicated that vitamin D status was related to the risk of ASD. The detection and appropriate intervention of vitamin D deficiency in ASD patients and pregnant and lactating women have clinical and public significance.

## 1. Introduction

Autism spectrum disorder (ASD) is a group of neurodevelopmental disorders characterized by impaired social interaction and communication, repetitive and stereotyped behaviors, and restricted interests [1]. Autism is becoming increasingly common. Recently, the Centers for Disease Control and Prevention reported that the prevalence of autism among 8-year-old children in the United States in 2016 was 1/54, with a 4.3:1 ratio of males to females [2]. Autistic individuals manifest problematic behaviors, such as attacking, self-injury, resistance to orders, and failure of normal conversation, and are usually comorbid with social-anxiety, attention-deficit/hyperactivity, sleep-wake disorders, and obsessive-compulsive disorders. As such, it is difficult for them to obtain the same education levels as their neurotypical peers, find full-time jobs, or live independently [1,3]. Fortunately, evidence indicates that appropriate and early intervention could help autistic individuals improve their symptoms and life quality [4]. Notably, research shows that nutritional and dietary intervention is an effective way to improve nutritional status, non-verbal IQ, and autism symptoms [5]. Therefore, it is essential to identify physiological dysfunction and abnormal nutritional status in autistic individuals and then take corresponding interventions.

Autism is a multifactorial disorder resulting from an interaction of genetic and environmental factors. Hundreds of autism risk genes and various environmental factors have been discovered [6,7]. Possible environmental factors include folic acid deficiency, neonatal hypoxia, maternal obesity, and gestational diabetes mellitus [7]. Recently, emerging evidence suggests that vitamin D deficiency might be an unfavorable factor of autism [8]. Vitamin D is a steroid hormone; it is primarily synthesized in the skin under UV-B light, and a small amount is derived from dietary intake [9]. Although the underlying mechanisms between vitamin D and autism are unclear, there is some biological evidence indicating the potential link. Vitamin D-metabolizing enzymes and vitamin D receptors are widely expressed in immune cells, the placenta, and the developing and adult brain; high levels of vitamin D surface receptor (protein disulfide isomerase family A members 3, PDIA3) are found in the cortex and hippocampus, which suggests the association between vitamin D, and brain development and function [10,11,12]. Indeed, vitamin D has important effects on brain development and function, including neuronal differentiation, proliferation and apoptosis, regulating synaptic plasticity, the ontogeny of the dopaminergic system, immunomodulation, and reducing oxidative burden [10]. In addition, vitamin D plays an important role in the regulation of gene expression. One study showed that 223 ASD risk genes in the SFARI database were vitamin D3-sensitive genes, which meant that these ASD related genes might be regulated by vitamin D [13].

Besides the plausible biological explanations, some epidemiological studies also reached related conclusions. A large number of case–control studies investigating the vitamin D status of children and adolescents with ASD from different countries and races showed that autistic children and adolescents had lower vitamin D status [14,15,16,17,18,19,20,21,22,23,24,25,26,27,28,29,30,31], but seven studies reached the opposite conclusions [32,33,34,35,36,37]. Moreover, several prospective studies investigated the role of maternal and neonatal vitamin D deficiency in autism onset. A nested case–control study from a Swedish population cohort suggested that neonatal vitamin D was slightly associated with a later risk of ASD, but maternal vitamin D was not related to ASD [38]. In contrast, a cohort study based on a Netherlands birth cohort revealed that neonatal vitamin D was not associated with ASD, but pregnant women with deficient vitamin D concentration at mid-gestation had a more than twofold chance to give birth to autistic infants [39]. Another nested case–control study from China supported the point that reduced neonatal vitamin D was associated with a higher risk of ASD [40], but studies from Canada and the United States did not support this point [41,42,43].

A meta-analysis on 11 case–control studies found that, compared with healthy children, children with ASD had 8.63 ng/mL lower 25(OH)D concentration overall [44]. However, as mentioned above, case–control studies measuring vitamin D levels of children and adolescents with ASD have constantly been emerging in recent years, with some contradictions in the results. In addition, several prospective studies have been performed on early-life vitamin D levels before ASD diagnosis, which provide evidence on whether autism onset is associated with reduced early-life vitamin D, also with inconsistent results. Individual vitamin D level is affected by many factors, including sunlight, diet, ethnicity, genetic polymorphism, and physiological conditions. Researchers could not completely control these factors between ASD and control groups, and the etiology and clinical manifestations of ASD are of high heterogeneity. Meta-analysis can merge the results of multiple studies and increase the sample size to reach a consistent conclusion and increase the credibility of that conclusion. Therefore, we conducted an updated meta-analysis of the case–control studies and meta-analysis of prospective studies to investigate links between vitamin D and ASD and to explore the potential source of heterogeneity between studies.

## 2. Materials and Methods

This systematic review and meta-analysis were conducted and reported in accordance with the Preferred Reporting Items for Systematic Reviews and Meta-Analyses (PRISMA) guidelines [45] and Meta-Analyses of Observational Studies in Epidemiology (MOOSE) guidelines [46]. The protocol was registered in the International Prospective Register of Systematic Reviews (PROSPERO) with a registration number of CRD42020161819 at www.crd.york.ac.uk/PROSPERO.

### 2.1. Search Strategy and Study Selection

We performed a systematic literature search in PubMed, EMBASE, Web of Science, and the Cochrane Library from database inception to 27 November 2019 to identify studies on vitamin D and the risk of autism. The search terms we used were MeSH phrases combined with text words relating to autism (“autism” OR “autistic” OR “ASD”) and vitamin D (“vitamin D” OR “1,25 dihydroxyvitamin d3” OR “d3,1,25 dihydroxyvitamin” OR “25 hydroxyvitamin d2” OR “25 hydroxyvitamin d3” OR “25(OH)D OR 1 alpha, 25 dihydroxy 20 epi vitamin d” OR “1,25 dihydroxy 20 epi vitamin d3”). No restrictions were applied for the languages, date, and location of the studies. Besides database searching, we manually checked the reference lists of the identified studies and relevant reviews. Two reviewers (WZQ and DR) independently checked the titles and abstracts of each paper to filter irrelevant papers and then read the full texts of the remaining studies to identify studies that met the eligibility criteria. Any disagreement was resolved by discussion.

### 2.2. Eligibility Criteria

Eligibility criteria were set according to PICOS approach: the participants (P), the interventions or exposure (I), the comparison (C), the outcome (O), the study design (S), as follows. Each letter in PICOS means a component: Participants: children or adolescents aged less than 18, pregnant women, and neonates. Intervention/exposure: insufficient or deficient vitamin D level in peripheral blood. Comparison: sufficient vitamin D level in peripheral blood. Outcome: autism spectrum disorder. Study design: case–control, cohort, and nested case–control studies. Duplicated studies with the same data were excluded. No limits were applied for the form of vitamin D. Studies had to report the mean and standard deviation of vitamin D concentration or odds ratio (OR)/relative risk (RR) for ASD incidence. Studies were excluded if participants were reported to be comorbid with any other disease that could affect vitamin D levels, such as epilepsy and ADHD.

### 2.3. Data Extraction

Data were independently extracted from eligible studies by two reviewers (WZQ and DR), including the first author’s name, year, country, study design, sample size, participants’ age, gender ratio, diagnostic criteria, sample for detecting, vitamin D measurement method, mean ± SD of vitamin D concentrations, *p*-value compared to controls, adjusted variable or confounding variable, and OR/RR (95% confidence interval (CI)) for ASD incidence.

### 2.4. Study Quality Assessment

For eligible studies, we used the Newcastle–Ottawa Scale (NOS) [47] to assess whether they had the general characteristics of an observational study. This scale comprised three aspects: study-participant selection, 0–4; the comparability of study participants, 0–3; the exposure or outcome of studies, 0–3, ranging from 0 to 9, among which 0–6 was regarded as low-quality, and 7–9 was high-quality. Two reviewers (WZQ and DR) independently evaluated the eligible studies. Any discrepancies were resolved by discussion.

### 2.5. Statistical Analysis

For the continuous variable, mean ± SD of vitamin D concentration was obtained to calculate the overall effect size and 95% confidence interval (95% CI). Mean difference (MD) was used to describe the difference of mean concentration between ASD and control groups in each study. If the concentration of vitamin D was present in nmol/L, we converted it into ng/mL, according to the formula, 1 ng/mL = 2.5 nmol/L. Meta-analysis of all eligible case–control studies was performed. In order to check the robustness of the results, we conducted an additional meta-analysis, barring data from previous meta-analyses [44]. In addition, prospective studies that reported mean ± SD were combined to examine if there was any difference in maternal and neonatal vitamin D between the ASD and control groups.

For the categorical variable, we obtained OR/RR from eligible studies to calculate the pooled OR/RR and 95% CI. Some case–control studies did not report vitamin D concentration but provided OR and 95% CI for the possibility of ASD exposed to vitamin D insufficiency or deficiency. Therefore, a meta-analysis on case–control studies with OR was performed. Prospective studies included nested case–control and cohort studies, providing either OR or RR. The prevalence of ASD was less than 10%, so we assumed that OR was approximatively equal to the RR and conducted meta-analysis from prospective studies.

Considering the anticipated large heterogeneity, we used the DerSimonian–Laird random-effects model for all meta-analyses. If there was no or low heterogeneity, the fixed-effects model was used. Cochran’s Q test and I^2^ statistic were used to measure heterogeneity. I^2^ referred to the percentage of heterogeneity, and I^2^ ≥ 50% indicated greater heterogeneity. Subgroup and meta-regression analyses were created to explore sources of heterogeneity. Subgroup analysis was based on the study population, measurement method, latitude, location, sample size, age, number of adjusted variables, and study quality. To eliminate the influence of individual studies, especially small sample and low-quality studies, leave-one-out sensitivity analysis was conducted. The funnel plot was used to evaluate publication bias, and Egger’s linear regression was conducted to check the symmetry of the funnel plot. A forest plot was created to visualize the overall effect size and 95% CI of the studies. All statistical analyses were performed in R (version 3.6.2), including the meta (version 4.11-0) and metaphor (version 2.1-0) packages. Two-tailed *p* < 0.05 was statistically significant.

## 3. Results

Figure 1 shows the process of identifying the eligible publications. A total of 1088 articles were retrieved through the databases: 221 records in PubMed, 439 records in EMBASE, 380 records in Web of Science, and 48 records in the Cochrane Library. In total, 16 articles were manually searched in the reference lists of relevant publications. After removing duplicates, 598 articles remained to be screened by title and abstract. Of these, 67 articles were left for full-text reading. Lastly, 34 eligible articles (a total of 20,580 participants) were included in the meta-analysis. Of these, 26 case–control studies [14,15,16,17,18,19,20,21,22,23,24,25,26,27,28,29,30,31,32,34,35,36,37,48,49,50] (1792 ASDs, 1969 controls) reported the blood vitamin D concentration of children and adolescents; three case–control studies [42,43,51], and two nested case–control studies [38,40](2687 ASDs, 3574 controls) examined the neonatal vitamin D concentration of participants; one case–control study [52] and one nested case–control study [38] (517 ASDs, 642 controls) assessed maternal vitamin D concentration of the ASD and control groups; two cohort studies [39,41] (5442 neonates, 3957 pregnant women) investigated the OR/RR for ASD incidence after being exposed to early-life vitamin D deficiency or insufficiency. The participants of two articles included not only neonates but also pregnant women, so there were 36 total studies from 34 articles.

The detailed study characteristics of each eligible study are demonstrated in Appendix A. In general, these studies were published from 2010 to 2019 and involved participants from Asia (*n* = 18), America (*n* = 6), Europe (*n* = 5), and Africa (*n* = 5). The ASD diagnostic criteria used in studies were DSM-Ⅳ, DSM-Ⅳ-TR, ADOS, ADIR, DSM-Ⅴ, ICD-9, ICD-10, ICD-F84.0, or a combination of the above. All eligible studies measured a total of 25(OH)D2 and 25(OH)D3 or 25(OH)D3 from serum, plasma, or dried blood spot as the biomarker of vitamin D; 25(OH)D3 was considered approximately equal to the total of 25(OH)D2 and 25(OH)D3, so the form of vitamin D in each study was not distinguished. The quality scores of the included studies are shown in Appendix A, ranging from 4 to 9, of which, 9 studies were evaluated as low-quality, 25 studies were high-quality, and the median NOS score of all studies was 7.

### 3.1. Meta-Analysis of Case–Control Studies Involving Children and Adolescents

A total of 24 case–control studies were included in the meta-analysis, providing mean ± SD vitamin D concentration in children and adolescents with and without ASD, of which two samples were plasma and 22 were serum. Meta-analysis showed that vitamin D concentration of the ASD group was 7.46 ng/mL lower than that of the control group (95% CI: −10.26; −4.66 ng/mL, *p* < 0.0001; Figure 2, Appendix A) using a random effects model, with a large heterogeneity (I^2^ = 98%, *p* < 0.01). In subgroup analysis, vitamin D measured by ELISA (MD: −10.19 ng/mL, 95% CI: −17.53; −2.86 ng/mL, *p* = 0.006) and radioimmunoassay (MD: −4.33, 95% CI: −6.81; −1.85, *p* = 0.0006) in the ASD group was significantly reduced compared to that of the control group, with slightly decreased heterogeneity between studies, while statistical significance disappeared in studies measured by HPLC (MD: −9.13, 95% CI: -19.33; 1.06; 1.06, *p* = 0.079) and LC–MS/MS (MD: −4.32, 95% CI: −15.20; 6.56, *p* = 0.436). With regard to latitude, subgroup analysis with a latitude below 30 (MD: −13.3, 95% CI: −20.83; −5.76, *p* = 0.0005) and between 30 and 40 (MD: −3.81, 95% CI: −5.83; −1.79, *p* = 0.0002) illustrated that vitamin D concentration was significantly lower in the ASD group than that in the control group. However, when latitude was beyond 40, there was no significant difference between the two groups. Studies performed in different areas showed quite different results. Subjects with ASD in Africa had largely reduced vitamin D concentration compared with that of the control group (MD: −15.56, 95% CI: −24.77; −6.35, *p* = 0.0009). In Asia, the difference in vitamin D concentration was also significant between the ASD and control groups (MD: −6.2, 95% CI: −9.15; −3.25, *p* < 0.0001). However, in Europe, subgroup analysis demonstrated that the ASD group had higher but nonsignificant vitamin D concentration than that of the control group (MD: 3.03, 95% CI: −6.78; 12.83, *p* = 0.545). In America, vitamin D levels did not differ between subjects with and without ASD (MD: −6.01, 95% CI: −13.42; 1.39, *p* = 0.11). More details about subgroup analysis are shown in Appendix A. Univariate meta-regression analysis indicated that latitude (*p* = 0.0107) was associated with a mean difference of vitamin D concentration between the two groups, accounting for 8.08% of heterogeneity.

Since 10 case–control studies reported OR, meta-analysis based on OR was conducted. Results indicated that reduced vitamin D status was significantly associated with increased risk of ASD (OR: 5.23, 95% CI: 3.13; 8.73, *p* < 0.0001, Figure 3, Appendix A). However, there was high heterogeneity between studies (I^2^ = 78.2%, *p* < 0.0001). The criteria for vitamin D deficiency or insufficiency were inconsistent. Included studies regarded 20 or 30 ng/mL as the cutoff of vitamin D insufficiency or deficiency. In the subgroup analysis, the association was significant when cutoff was 30 ng/mL (OR: 6.13, 95% CI: 3.39; 11.09, *p* < 0.0001, I^2^ = 83.7%), but association was nonsignificant when cutoff was 20 ng/mL (OR: 2.83, 95% CI: 0.91; 8.72, *p* = 0.07, I^2^ = 40.2%). On the basis of latitude, assessment methods, age, study quality, and number of adjusted variables, all subgroups demonstrated a significant association between reduced vitamin D status and increased risk of ASD (Appendix A).

### 3.2. Meta-Analysis of Case–Control Studies Barring Data from Previous Meta-Analysis

Previous meta-analysis integrated case–control studies measured vitamin D concentration in ASD and control groups before May 2015. In order to evaluate the robustness of the association between ASD and vitamin D status, we carried out meta-analysis of 15 studies excluding studies before May 2015. Results showed that children and adolescents with ASD had 6.2 ng/mL lower vitamin D concentration than that of the control group (95% CI: −9.62; −2.78, *p* = 0.0004, I^2^ = 96.8%; Figure 4, Appendix A), which was similar to results of the previous meta-analysis [44]. Significant difference in vitamin D status between the two groups was observed in several subgroups: latitude between 30 and 40, mean age of participants >5, high study quality, adjusted variable = 2 (Appendix A). Univariate meta-regression analysis suggested that age (*p* = 0.0486) had a slightly significant effect on the mean difference of vitamin D concentration between the two groups.

### 3.3. Meta-Analysis of Prospective Studies about Neonates and Pregnant Women

Meta-analysis of maternal and neonatal vitamin D concentration indicated that there was a trend of lower vitamin D concentration in subjects with ASD (MD: −3.15, 95% CI: −6.57; 0.26, *p* = 0.07, I^2^ = 99%; Figure 5, Appendix A). Subgroup analysis showed that maternal and neonatal vitamin D concentration in the ASD group was 3.04 and 3.28 ng/mL lower than that of the control group (95% CI: −6.86; 0.77, *p* = 0.102, I^2^ = 93.3% and 95% CI: −8.47; 1.91, *p* = 0.228, I^2^ = 99.3%), respectively, but no statistical significance was observed.

Nested case–control and cohort studies were also summarized to a pooled OR of 1.54 (95% CI: 1.12; 2.10; Figure 6, Appendix A), suggesting that the lower level of maternal and neonatal vitamin D caused a 54% higher risk of later ASD onset. Subgroup analysis demonstrated that decreased maternal vitamin D concentration contributed to the development of ASD (OR: 2.72, 95% CI: 1.61; 4.59, *p* = 0.0002, I^2^ = 44.7%) but neonatal vitamin D concentration was not found to be significantly related to the risk of ASD (OR: 1.2, 95% CI: 0.94; 1.54, *p* = 0.15, I^2^ = 65.3%).

### 3.4. Sensitivity Analysis and Publication Bias

Leave-one-out sensitivity analysis was applied to check if there was any individual study affecting the overall results. For three separate meta-analyses of case–control studies, we did not find any outlier that significantly influenced the results. Intriguingly, with respect to meta-analysis on maternal and neonatal vitamin D concentrations, sensitivity analysis suggested that the elimination of Wu et al. (2017) [40] (MD: −1.43, 95% CI: −2.63; −0.24, *p* = 0.0189, I^2^ = 83.3%) or Windham et al. (2019) [43] (MD: −3.79, 95% CI: −7.58; −0.002, *p* = 0.0499, I^2^ = 99.1%), respectively, led to a significant difference between the ASD and control groups. Sensitivity analysis of the meta-analysis of prospective studies did not significantly alter the summarized results. The funnel plots shown in Appendix A indicate that all meta-analyses have no publication bias, and *p* > 0.05 in both the Egger’s test and Begg’s test.

## 4. Discussion

The present meta-analysis confirmed that children and adolescents with ASD have significantly lower vitamin D concentration than that of healthy children and adolescents, which was consistent with previous meta-analysis [44]. Both meta-analyses of the ORs in 10 case–control studies and of vitamin D concentrations in 15 case–control studies conducted after May 2015 yielded the same findings, which increased the credibility of the results. Furthermore, overall estimates of vitamin D concentrations in prospective studies indicated that early-life vitamin D levels of both maternal and neonatal vitamin D tended to be lower in subjects later diagnosed with ASD. Meta-analysis of ORs in prospective studies showed that decreased early-life vitamin D led to a 54% higher risk of later diagnosed ASD. In subgroup analysis, maternal vitamin D was shown to be associated with ASD, but neonatal vitamin D was not.

There are several probable reasons for the phenomenon that children and adolescents with ASD have lower vitamin D concentration than that of healthy controls. First, the lifestyle habits of ASD children are different from healthy children. Compared with healthy children, ASD children are pickier eaters, eating limited kinds of foods and consuming less vitamin D [53]. Moreover, one study showed that ASD children spent less time on outdoor activities than healthy controls did in the second year of life, which suggested that autistic children were less exposed to solar UV-B, indicating that they received less vitamin D from cutaneous synthesis [54]. These factors may be partly responsible for the lower vitamin D status in ASD children. Second, vitamin D levels may be related to genetic factors. Vitamin D metabolic and vitamin D receptor gene variants that were shown to be associated with ASD risk might influence vitamin D status [55,56,57]. The use of drugs like antiepileptic drugs might also cause vitamin D loss [58].

In the subgroup analysis, we found that case–control studies from different areas yielded different estimates. Studies from African countries were estimated to show maximal vitamin D concentration difference between ASD and control groups, followed by Asian countries; estimates from American and European countries were not significant. This may be due to different health care services and awareness of autism management. Furthermore, latitude was associated with mean difference according to the meta-regression, although latitude can only account for 8% heterogeneity. Subgroup analysis stratified by latitude showed that participants with and without ASD from lower-latitude areas presented larger vitamin D concentration differences. Studies from the latitude and area subgroups were little overlapped. Data from these meta-analyses showed the mean vitamin D level of children with and without ASD from low-latitude areas was higher than that from high and medium latitude areas, respectively, which suggested that latitude was an important factor influencing vitamin D level. However, the difference in vitamin D concentration between children with and without ASD in low-latitude areas became larger; that is to say, increasing light exposure in autistic children did not cause an equal increase in vitamin D as in the healthy controls. Therefore, we hypothesized that children with ASD might show a weak ability of cutaneous synthesis of vitamin D, which remains to be investigated.

HPLC and LC–MS/MS are gold standards for measuring vitamin D concentration, but the vitamin D level between ASD and control groups was not significantly different in the subgroup analysis of these two methods. Considering the special status of HPLC and LC–MS/MS, we combined these two subgroups for additional meta-analysis and found that vitamin D concentration in the ASD group was lower than that in the control group (MD: −6.72, 95% CI: −13.99; 0.55, *p* = 0.07, I^2^ = 97.8%), but it was still not statistically significant. These two subgroups consisted of six studies, of which the results of three studies were significant (*p* < 0.05) and the other three were not (*p* > 0.05). The average vitamin D concentration of the ASD groups in the six studies was in the range of 10–30 ng/mL (Appendix A), which meant that children with ASD were with low vitamin D status. With regard to the control groups, average vitamin D concentration in three studies with significant results was above 30 ng/mL (Appendix A), but in the three studies with nonsignificant results, it was in the range of 10–30 ng/mL (Appendix A), which was comparable to the ASD group. Vitamin D deficiency or insufficiency are global problems. Thus, we speculated that the reason why there was no significant difference in vitamin D concentration between the ASD and control groups was because the control children were also in low vitamin D states.

Vitamin D status after ASD diagnosis is affected by lifestyle characteristics, such as diet and outdoor activities. Fortunately, early-life vitamin D status is not affected by children’s lifestyle factors. Vitamin D can be transferred to the fetus through the placenta to support fetal development, so a fetus’ vitamin D status depends on maternal vitamin D concentration. However, nearly one-half of pregnant women lack vitamin D [59]. The neonatal period is also a sensitive period of neurodevelopment, and vitamin D plays a pivotal role in this period. Vitamin D deficiency during neurodevelopmental periods could result in brain-structure alterations and behavioral problems [60,61]. Thus, we combined maternal and neonatal vitamin D studies to determine whether their insufficiency or deficiency was associated with ASD diagnosis. Our results showed that early-life vitamin D deficiency led to a slight increase in ASD risk. However, there were some points about cutoffs to distinguishing vitamin D status. First, cutoffs in the studies were different; some chose 20 ng/mL as the criterion [62], while others chose the lowest quantile as vitamin D deficiency. Second, neonatal vitamin D was lower than maternal vitamin D, but researchers used the same standard to define maternal and neonatal vitamin D deficiency. Different cutoffs might exert an influence on the pooled estimates. On the other hand, vitamin D was measured at a single time and no other time-point in all studies, which could not represent the average level of vitamin D concentration during developmental stages. At this point, we can only speculate whether early-life vitamin D is linked to ASD risk. Future large-scale birth cohort studies and well-designed randomized controlled-trial studies about vitamin D supplementation effects of ASD are needed to confirm this association.

Several underlying mechanisms may explain the association between vitamin D deficiency and ASD. A large number of studies showed that vitamin D significantly contributes to neurodevelopment, playing important roles in neurogenesis, cell proliferation, differentiation, apoptosis, and neurotransmitter metabolism [63]. Vitamin D also has anti-inflammatory and antioxidant properties; for instance, vitamin D supplementation decreased serum interleukin 10 and 12 concentration and increased total antioxidant capacity [64]. In addition, one study showed that vitamin D deficiency induced increased reactive oxygen species (ROS) in the periphery and brain and caused excitatory and inhibitory neurotransmitter imbalance in animal models [65]. Autism is regarded as a condition that affects the brain, and increasing evidence has indicated that oxidative stress and inflammation are involved in the pathogenesis of autism, which may be related to vitamin D deficiency.

### 4.1. Implication

Our findings indicated that vitamin D level in children and adolescents with autism is significantly lower than that in healthy controls, which has clinical implications. Considering the importance of vitamin D and the high prevalence of vitamin D deficiency, regular screening of vitamin D levels in autistic individuals and necessary intervention are recommended. Furthermore, pregnant and lactating women consume more vitamin D than usual and are generally deficient in vitamin D [66]; maternal and neonatal vitamin D status may be associated with subsequent diagnosis of ASD. Vitamin D status should be included in routine screening during pregnancy and lactation in order to provide appropriate clinical intervention.

### 4.2. Limitations

Several limitations in this study should be considered. First, the causal relationship between vitamin D and autism could not be confirmed. Case–control studies of vitamin D levels in autistic individuals cannot provide evidence of causation. In the prospective studies, researchers only measured vitamin D levels at one point, which may have failed to reflect vitamin D status across developmental stages. Second, large heterogeneity was observed across studies, which may have resulted from differences in demographic characteristics, measurement methods of vitamin D, seasons, and adjusted variables. Although a random-effects model was used, a substantial amount of heterogeneity remained. Therefore, the effect sizes of meta-analysis should be interpreted with caution. However, a significant relationship between vitamin D status and autism persisted in most subgroups stratified by multiple study characteristics. Third, different cutoffs were chosen in the prospective studies to define vitamin D deficiency, under which circumstances, the same vitamin D level may have belonged to different categories. In addition, in the same study, researchers applied the same kind of standard to determine maternal and neonatal vitamin D status, which led to bias because maternal vitamin D concentration is much higher than neonatal vitamin D concentration is.

## 5. Conclusions

Our findings suggest that vitamin D status has an association with autism. Some caution should be taken when results are interpreted because of the substantial heterogeneity between studies and unconfirmed causality. Given the high prevalence of vitamin D deficiency in children and adolescents with autism and pregnant and lactating women, screening and appropriate interventions for vitamin D may have significant effects on autism prevention and treatment. Further studies are needed to investigate the causal relationship between vitamin D and autism and elucidate its mechanism.

## Figures and Tables

**Figure 1 nutrients-13-00086-f001:**
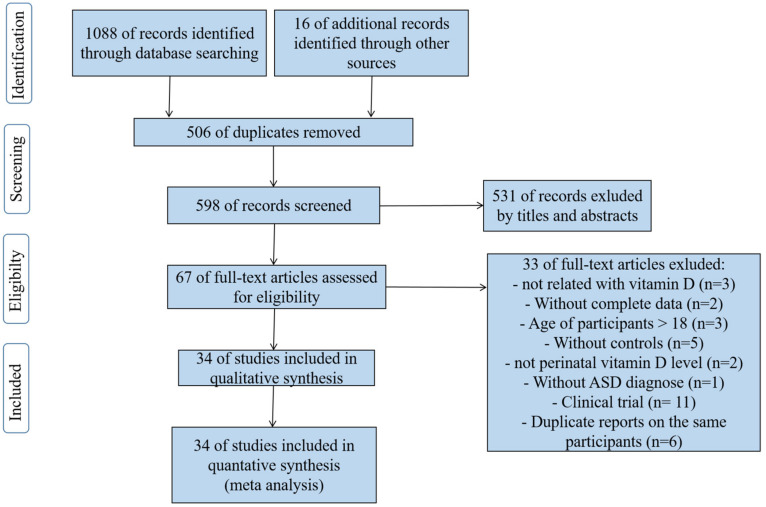
Flow chart of identification of eligible studies.

**Figure 2 nutrients-13-00086-f002:**
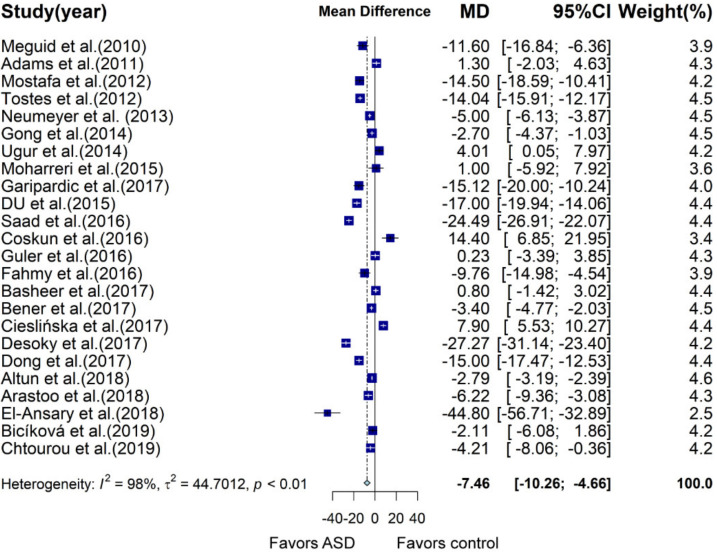
Forest plot of meta-analysis of 24 case control studies based on vitamin D concentration, showing that children and adolescents with autism spectrum disorder (ASD) have an average of 7.46 ng/mL lower vitamin D concentration than that of the controls, with 98% heterogeneity. MD, mean difference; CI, confidence interval.

**Figure 3 nutrients-13-00086-f003:**
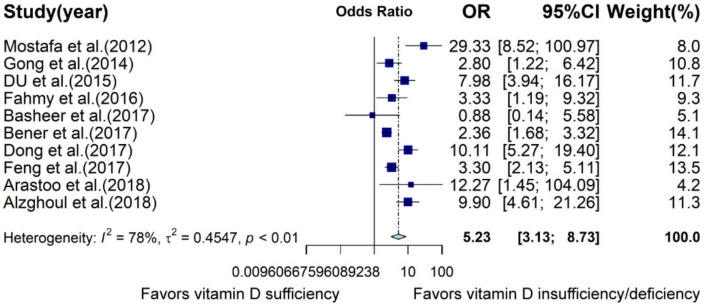
Forest plot of meta-analysis of 10 case–control studies based on odds ratio (OR), demonstrating that children with vitamin D insufficiency and deficiency are about 5.23 times more likely to develop ASD than vitamin D sufficient children are, with 78% heterogeneity. CI, confidence interval.

**Figure 4 nutrients-13-00086-f004:**
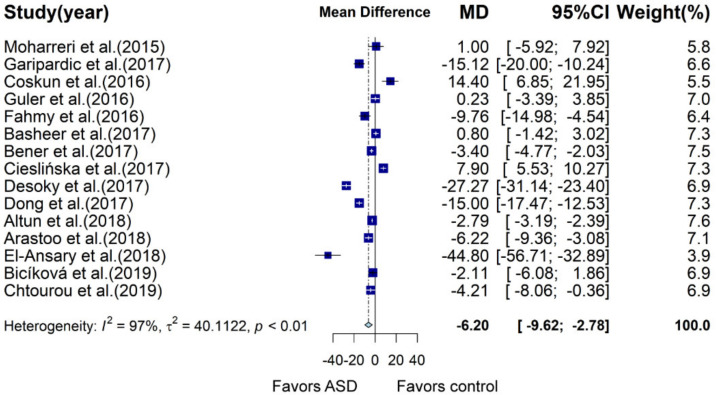
Forest plot of meta-analysis of 15 case–control studies based on vitamin D concentration, showing that children and adolescents with ASD have an average of 6.20 ng/mL lower vitamin D concentration than that of the control in studies conducted after May 2015, with 97% heterogeneity. MD, mean difference; CI, confidence interval.

**Figure 5 nutrients-13-00086-f005:**
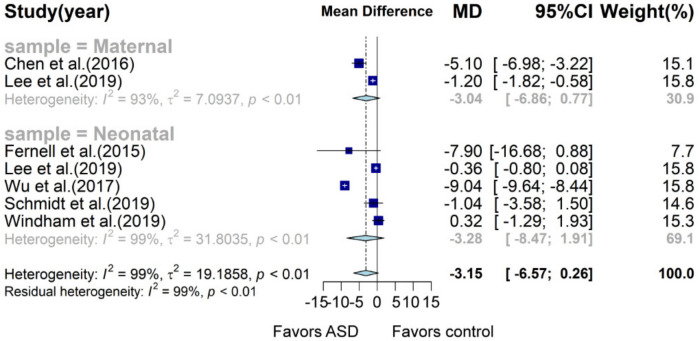
Forest plot of meta-analysis of seven studies based on vitamin D concentration. Overall, children with ASD tend to have 3.15 ng/mL lower neonatal or maternal vitamin D concentration than that of children without ASD, with 97% heterogeneity. MD, mean difference; CI, confidence interval.

**Figure 6 nutrients-13-00086-f006:**
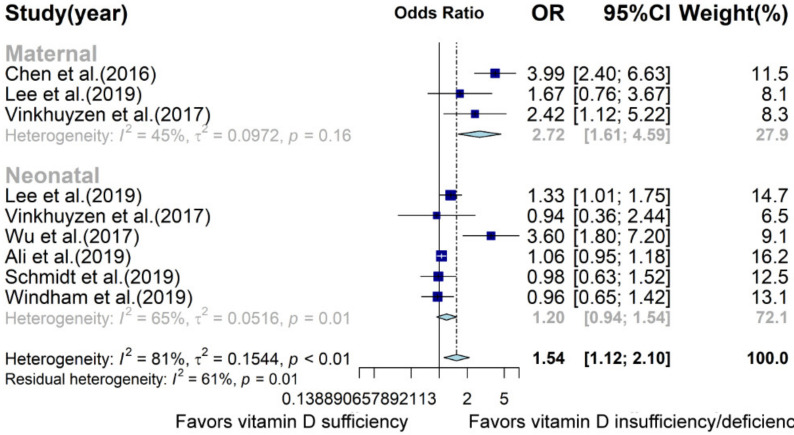
Forest plot of meta-analysis of nine prospective studies based on OR. Overall, children with lower maternal or neonatal vitamin D levels have a 54% higher chance to develop ASD, with 81% heterogeneity. OR, odds ratio; CI, confidence interval.

## Data Availability

All data obtained from published papers. Source PubMed, EMBASE, Web of Science and the Cochrane Library.

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
