# Peer review of "The Association between Vitamin D Status and Autism Spectrum Disorder (ASD): A Systematic Review and Meta-Analysis"

_nutrients, 2020, doi:10.3390/nu13010086_

Round 1

Reviewer 1 Report

The research described in this study duplicates previous work in this field. As the authors have stated there have been conflicting studies - but the authors do not explain why in previous studies there were conflicting results.

The authors also state that in studies that used HPLC statistical significance disappeared (lines 202-204). Given that the gold standard of vit D measurement is HPLC/mass spec (Liquid-chromatography-tandem-mass-spectrometry (LC-MS/MS)) I think this deserves much greater consideration - instead it is simply combined with all other studies in their analysis.

Lastly, and most importantly, the authors definitions of vitamin D deficiency and insufficiency are incorrect.

In the UK vitamin D deficiency is defined as less than 30 nmol/L serum 25(OH)D, and insufficiency is defined as levels in the range of 25–50 nmol/L may be inadequate for some people; Serum 25(OH)D levels greater than 50 nmol/L are sufficient level for most people. 

There is no international consensus on what constitutes deficiency but following are levels from two different countries - the USA and China.

After reviewing data on vitamin D needs, an expert committee of the Food and Nutrition Board (FNB) at the National Academies of Sciences, Engineering, and Medicine (NASEM) concluded that people are at risk of vitamin D deficiency at serum 25(OH)D concentrations less than 30 nmol/L (12 ng/mL; see Table 1 for definitions of “deficiency” and “inadequacy”) [1]. Some people are potentially at risk of inadequacy at 30 to 50 nmol/L (12–20 ng/mL). Levels of 50 nmol/L (20 ng/mL) or more are sufficient for most people. In contrast, the Endocrine Society stated that, for clinical practice, a serum 25(OH)D concentration of more than 75 nmol/L (30 ng/mL) is necessary to maximize the effect of vitamin D on calcium, bone, and muscle metabolism [11,12]. The FNB committee also noted that serum concentrations greater than 125 nmol/L (50 ng/mL) can be associated with adverse effects [1] (Table 1).  

And in China, the Osteoporosis Committee of China Gerontological Society adopted the same standard as the National Academy of Medicine, with vitamin D deficiency defined as less than 30 nmol/L, insufficiency as 30–49.9 nmol/L, and sufficiency as more than 50 nmol/L9.

In this paper however the authors have defined vitamin D deficiency as 25 OH vitamin D concentrations of < 20 NANOGRAMS/ml - which is equivalent to 50 nmol/L - no country that I am aware of defines deficiency as < 50 nmol/L. The same applies to their definition of insufficiency.

Reviewer 2 Report

The manuscript entitled „The association between vitamin D status and autism spectrum disorder (ASD): a systematic review and meta-analysis” presents the meta-analysis results for vitamin D status in correlation to ASD.

In general publication is interesting, and subject is important.The manuscript consists of 6 figures, and 66 references. I had few questions about limitations, but section „Limitation” is well prepared.

Data on vitamin D level are varied, and it is influenced by, among others. measurement season, diet or supplementation. Accordingly, it is not always possible to obtain reliable results. Unfortunatelly I had no access to supplementary materials, so I believe that data present in this section are more informative.

Reviewer 3 Report

Wang et al. conduct an update meta-analysis of case-control studies and a meta-analysis of prospective studies to investigate links between vitamin D and ASD.

Their meta-analysis of 24 case-control studies demonstrates that children and adolescents with ASD have a significant lower vitamin D concentration than control group. Furthermore, a quantitative integration of 10 case-control studies reporting OR reveals lower vitamin D is associated with higher risk of ASD. Finally, meta-analysis of maternal and neonatal vitamin D reports a trend of decreased early life vitamin D concentration in ASD group.

Authors performed a good work, also explaining the plausible mechanisms related to the observed results and the limitation that are often present in all studies evaluating relation between vitamin D.

1. only minor english editing

Author Response

Dear reviewer,

Thanks very much for taking your time to review this manuscript (Title: The association between vitamin D status and autism spectrum disorder (ASD): a systematic review and meta-analysis; Manuscript ID: nutrients-1025505).

Thanks for your approval of our work, which has inspired us to further investigate the etiology of ASD. We also attached great importance to your comment about English editing. We used English editing service provided by MDPI, so our manuscript was edited by an English native speaker. The highlights scattered in the manuscript have been edited.

Best wishes,

Authors of this manuscript

Reviewer 4 Report

This meta analysis is well performed and informative,

even though some limiations which were alreday described.

Thank you for your contributions.

Author Response

Dear reviewer,

Thanks very much for taking your time to review this manuscript (Title: The association between vitamin D status and autism spectrum disorder (ASD): a systematic review and meta-analysis; Manuscript ID: nutrients-1025505).

Thanks for your approval of our work. We did spend some time to conduct this research, and we hope this work will contribute to the field. We will also continue to explore the relationship between ASD and vitamin D and make efforts to discover the cause, prevention and treatment of ASD.

Best wishes,

Authors of this manuscript